# Inherent strain and kinetic coupling determine the kinetics of ammonia synthesis over Ru nanoparticles

Yuqi Yang ✉, Anders Hellman ✉ & Henrik Grönbeck ✉

The large-scale ammonia synthesis using the Haber-Bosch process is crucial in modern society and the reaction is known to be facile over Ru-based catalysts. Herein, first-principles kinetic Monte Carlo (kMC) simulations are utilized to explore the reaction kinetics on Ru nanoparticles (NPs), extending the current knowledge that is mainly based on calculations of single crystal surfaces. It is only by accounting for the effects of kinetic couplings between different sites and inherent strain in the NPs that experimental turnover frequencies (TOFs) can be reproduced. The enhanced activity of inherently strained NPs is attributed to the co-existence of sites with both tensile and compressive strain, which simultaneously promotes $N_2$ dissociation and $NH_x$ (x = 0, 1 and 2) hydrogenation. We propose that kinetic couplings on Ru NPs with tailored strain-patterns offer a strategy to break the limitations of linear scaling relations in the design of ammonia synthesis catalysts.

The Haber-Bosch process for ammonia synthesis from $N_2$ and $H_2$ is one of the most important scientific inventions in modern society[1,2]. Ammonia is used in the production of fertilizers or as an energy carrier for use as fuel or in power plants[2]. The Haber-Bosch process requires harsh reaction conditions with temperatures in the range of 673 –773 K and high pressures (10–30 MPa). In fact, nearly 2% of the world's energy consumption is attributed to the Haber-Bosch process[3], which motivates the search for ammonia synthesis catalysts with activity at lower temperatures. Ru is known to be the ideal metal catalyst for ammonia synthesis under industrial conditions[4–7]. Ru is a scarce metal, and to utilize the catalyst efficiently and sustainably, there is a need to develop a new generation of Ru catalysts with higher activity[8–10].

Technical Ru catalysts for the ammonia synthesis reaction are composed of supported Ru nanoparticles (NPs), exposing different crystal facets and sites to the gas phase. The sizes and shapes of Ru NPs are generally ill-defined, making the contributions of different particles to the kinetic properties difficult to disentangle. Thus, it is challenging to establish clear structure-reactivity relations for technical Ru catalysts. Experimentally, fundamental studies are typically carried out on well-defined Ru surfaces to facilitate the understanding of the kinetic importance of different types of sites[11–14]. Computationally, the

potential energy surfaces for ammonia synthesis over different single-crystal surfaces have been evaluated using density functional theory (DFT) calculations[8,15]. Moreover, different types of DFT-based mean-field microkinetic models have been constructed to study the kinetic behavior and identify active sites. The present understanding is that $N_2$ dissociation is limiting the rate and that this elementary reaction is facilitated by the presence of step sites, such as the well-known $B_5$ site[8]. Following the knowledge of active sites, strategies have been proposed to promote the reactivity of undercoordinated sites[16] or to increase the number of active sites on Ru NPs[17–19].

Modeling catalytic NPs as isolated single-crystal surfaces does not capture the structural complexity of NPs, which contain a range of different sites. The site assembly offers the possibility of kinetic couplings where different parts of the reaction occur on different sites according to the principle of least resistance[20–22]. Moreover, model catalysts with ideal surfaces generally do not account for the inherent strain, which is always present in metal NPs. Technical NPs are inherently strained due to the structural mismatch caused by grain boundaries, interaction with the support material, and surface defects[23–25]. Strain is known to affect the d-states of Ru[26], which modifies the potential energy surface of the ammonia synthesis

Department of Physics and Competence Centre for Catalysis, Chalmers University of Technology, Göteborg, Sweden. ✉e-mail: yuqiy@chalmers.se; ahell@chalmers.se; ghj@chalmers.se

reaction[27,28]. As a consequence, the surface models commonly used in kinetic analysis of the ammonia synthesis reaction over Ru make it challenging to describe kinetic properties, such as turnover frequency (TOF) and apparent activation energy of technical Ru NPs.

The recent development of advanced synthesis and characterization techniques for well-defined Ru NPs opens an opportunity to link the structure and function of technical Ru NPs. High-resolution transmission electron microscopy makes it possible to measure the strain distribution in Ru NPs directly[29–31]. The influence of kinetic couplings and inherent strain on catalytic activity is, however, still difficult to uncover solely based on experiments. In this work, first-principles based kinetic Monte Carlo (kMC) simulations are performed to investigate the kinetic properties of Ru NPs for ammonia synthesis. We find that the combined effect of structural complexity and inherent strain is required to account for the experimentally measured TOF. The simulations reveal that kinetic couplings between different sites yield high catalytic activity, as the $N_2$ dissociation and $NH_x$ (x = 0, 1, and 2) hydrogenation steps occur on different sites, which communicate via rapid adsorbate diffusion. Inherent random strain further enhances the activity by promoting the reaction on sites for both $N_2$ dissociation and $NH_x$ hydrogenation. Our work suggests that kinetic coupling on strained-designed Ru NPs offers a possibility to break the limitation of scaling relations[32] in ammonia synthesis, thereby enabling the design of new generations of Ru catalysts.

## Results and discussion
### Reaction energy landscapes
Ammonia synthesis over metal catalysts is known to follow the dissociative reaction mechanism, where the first step is $N_2$ dissociation into atomic N, followed by sequential hydrogenation of $NH_x$ to $NH_3$[33]. The seven elementary reaction steps considered in this work are:

$$N_2(g) + * \rightleftharpoons N_2{}^* \tag{1}$$

$$N_2{}^* + {}^* \rightleftharpoons 2N^* \tag{2}$$

$$H_2(g) + 2\,{}^* \rightleftharpoons 2H^* \tag{3}$$

$$N^* + H^* \rightleftharpoons NH^* + {}^* \tag{4}$$

$$NH^* + H^* \rightleftharpoons NH_2{}^* + {}^* \tag{5}$$

$$NH_2{}^* + H^* \rightleftharpoons NH_3{}^* + {}^* \tag{6}$$

$$NH_3{}^* \rightarrow NH_3(g) + {}^* \tag{7}$$

Gaseous species are labeled with (g), and an asterisk represents a surface site.

The potential energy surface for the reaction over Ru NP is complex as the different sites have unique adsorption and reaction energies. Here, the potential energy surface is described by scaling relations using the site-specific generalized coordination number (GCN), see Fig. 1a. The linear relations are fitted to DFT calculations considering each surface species in the stable adsorption sites on different surfaces, with respect to gaseous $N_2$ and $H_2$, see Supplementary Tables 1–6. The positive slopes mean that the low GCN sites have higher adsorption energies. Thus, the surface species formed on high GCN sites tend to diffuse to sites with low GCN. The adsorption energy of N on Ru(0001) is an outlier and deviates from the linear scaling relation (solid mark at GCN = 7.5). As Ru(0001) is an important reference system, we use the explicit DFT-calculated adsorption

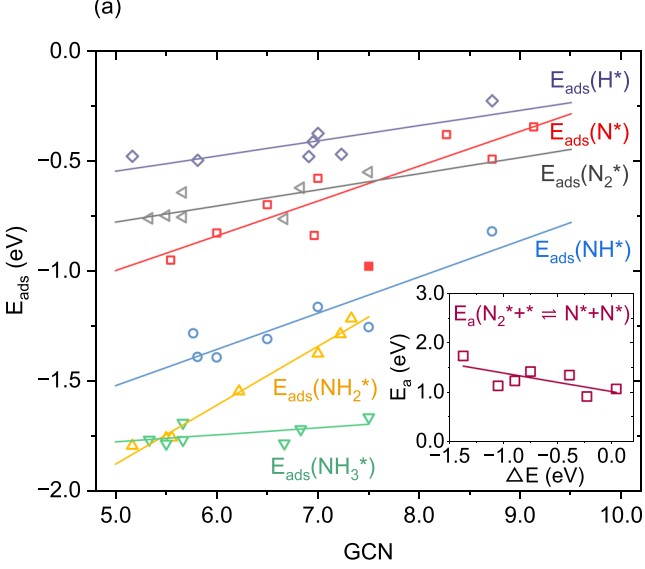

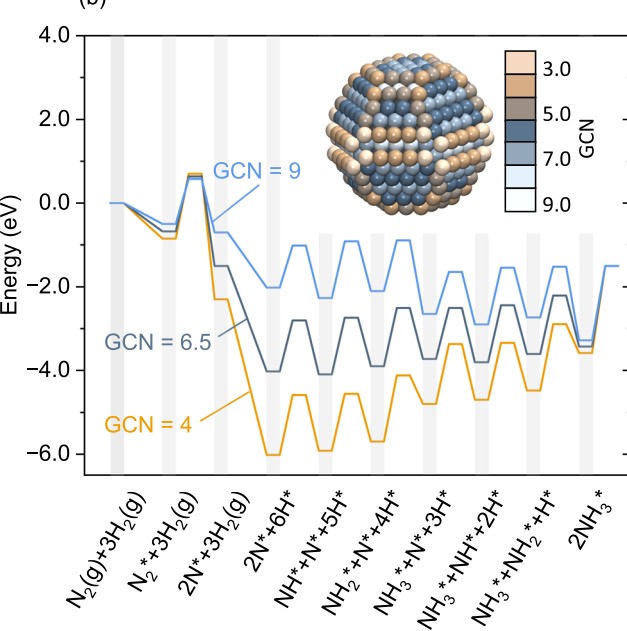

**Fig. 1 | Construction of potential energy landscapes for $NH_3$ synthesis on Ru. a** Adsorption energies versus Generalized Coordination Number (GCN) for each surface species (The data points are shown in Supplementary Tables 1–6.) Inset: energy barriers of $N_2$ dissociation step versus reaction energy. **b** Potential energy surfaces of sites with GCN of 9, 6.5, and 4. The vertical shadings link the potential energy surface to the intermediates. Inset: the $Ru_{780}$ NP colored according to the GCN of atop sites.

energy of the N atom in the kinetic simulations of the extended surface. Importantly, sites with GCN = 7.5 are minority sites on the Ru NPs, with a low contribution to the overall TOF (see below). Linear scaling relations are used to evaluate the transition state energies as a function of the reaction energies. The scaling relation for $N_2$ dissociation has a slope of −0.38, which means that a high reaction energy corresponds to a low energy barrier (inset Fig. 1a). For the $NH_x$ hydrogenation steps (see Supplementary Fig. 1), the slopes are 1.23, 0.15 and 0.26 for N, NH and $NH_2$ hydrogenation, respectively.

The thermodynamic equilibrium shape of Ru NPs is given by a Wulff construction based on the surface energies of Ru(0001), Ru(10$\bar{1}$0), Ru(10$\bar{1}$1), Ru(10$\bar{1}$2) and Ru(2$\bar{1}$$\bar{1}$2) (see Supplementary Table 8). In Fig. 1b, the considered $Ru_{780}$ NP with a size of 2.68 nm is

colored according to the GCN of atop sites. On Ru₇₈₀, the GCN takes values between 3.67 and 9.17. The wide GCN range offers different environments for each elementary reaction during ammonia synthesis. For Ru₇₈₀, the sites with GCN lower than 6 are mainly located on edges and corners, GCN between 6 and 8 are located on Ru(0001) and Ru(10$\bar{1}$1) facets, whereas GCN higher than 8 are located on steps and partly on Ru(10$\bar{1}$0) facets. Note that the $B_5$ sites (GCN of 8.75) are not present on Ru₇₈₀, although $A_5$ sites with a GCN of 9.17 are present.

Figure 1b shows the potential energy surfaces as obtained from the linear scaling relations for GCN of 9, 6.5, and 4, which represent the energy properties of edge, facet, and step sites, respectively. The sites with a high GCN are associated with a low transition state energy for the $N_2$ dissociation step. This observation is consistent with previous reports, suggesting that step sites with high GCN are the active sites for $N_2$ dissociation[8,12]. The potential energy surfaces derived from linear scaling relations underestimate the difference in the $N_2$ dissociation barrier over Ru(0001) and a stepped Ru surface. In the explicit DFT calculations, the $N_2$ dissociation energy barriers with respect to gaseous $N_2$ on Ru(0001) and stepped Ru(0001) surfaces are calculated to be 1.18 and 0.16 eV, respectively. The calculated difference of 1.02 eV is consistent with the experimental value of 0.9 eV[12]. In comparison with the explicit DFT calculations, the energy barrier for $N_2$ dissociation derived from the linear scaling relationships is underestimated by 0.26 eV on the Ru(0001) surface, whereas it is overestimated by 0.42 eV on the stepped Ru(0001) surface, yielding a difference in barriers of only 0.34 eV. For the subsequent $NH_x$ hydrogenation steps, the difference in N and H adsorption energies is more significant than the difference in $NH_x$ hydrogenation energy barriers. This suggests that N and H adsorption energies are important when identifying the active sites for the $NH_x$ hydrogenation steps. By comparing the N and H adsorption energies on different sites, the $NH_x$ hydrogenation steps will preferentially occur on the low GCN sites, which is different from the $N_2$ dissociation step. The inspection of the potential energy surfaces, thus, suggests a separation of preferred sites for the $N_2$ dissociation and $NH_x$ hydrogenation steps.

## Kinetics of ammonia synthesis over Ru nanocatalysts

Kinetic simulations are performed using the potential energy landscapes obtained from the linear scaling relations. Sites are allowed to communicate via adsorbate diffusion, and TOF is defined as the number of formed gas-phase $NH_3$ molecules per surface site and second. To confirm the performance of parametrized energy landscapes, we simulate first ammonia synthesis over the Ru(10$\bar{1}$3) surface, which allows for comparison with recent operando experiments[14]. At the same reaction conditions, we obtain coverages of N, NH, and $NH_2$ that are consistent with experiments (see Supplementary Table 9).

The simulated TOFs of different Ru systems are compared to experiments in Fig. 2a. Experimentally, the TOF of Ru NPs on inert supports ranges from 0.002 site⁻¹ s⁻¹ to 0.04 site⁻¹ s⁻¹ (see Supplementary Table 10 for details on the experimental reaction conditions and TOFs). The simulations are performed at a temperature of 673 K and partial $N_2$ and $H_2$ pressures of 2.5 bar and 7.5 bar, respectively. The TOFs on Ru(0001) and stepped Ru surfaces with $B_5$ sites are calculated to be $3.7 \times 10^{-6}$ and $7.0 \times 10^{-5}$ site⁻¹ s⁻¹, respectively. The higher TOF on the stepped surface confirms the promotion effect of $B_5$ sites on the activity. The ratio between the TOFs for Ru(0001) and the stepped surface is only 20, which is substantially smaller than the difference of about 5 orders of magnitude observed experimentally[12]. When using the explicit DFT potential energy surfaces in the kMC simulations, the TOFs for Ru(0001) and the stepped Ru surface are determined to be $6.3 \times 10^{-9}$ and $3.8 \times 10^{-4}$ site⁻¹ s⁻¹, respectively. Thus, the TOF over the stepped surface is about $6 \times 10^4$ times higher than the TOF on the Ru(0001) surface, which is consistent with the experiments. Comparing the TOFs calculated by the explicit potential energy surface and by the linear scaling relations, the errors in $N_2$ dissociation energy barriers

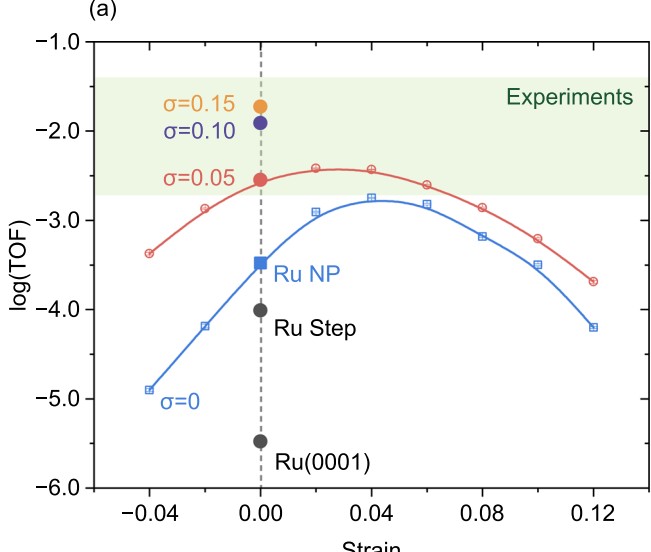

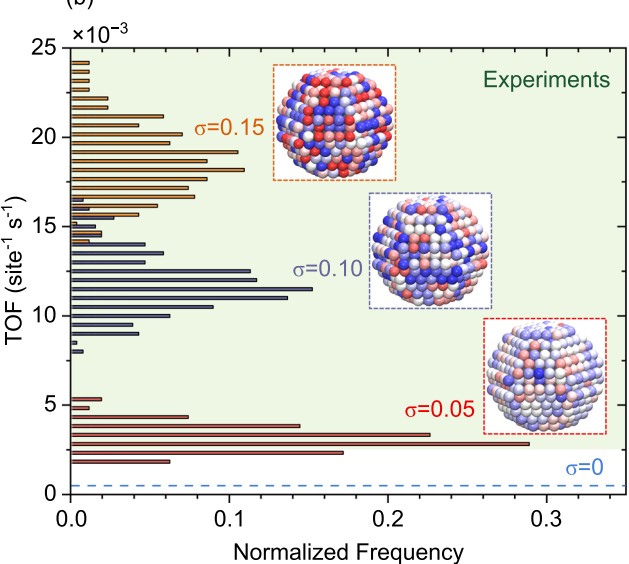

**Fig. 2 | TOF for $NH_3$ synthesis over Ru. a** Calculated TOFs on Ru surfaces and NPs versus strain. The error bars are smaller than the data marks and are calculated as the standard deviation of 16 simulations. **b** TOF distributions for 256 different randomly strained Ru NPs at $\sigma$ values of 0.05, 0.10, and 0.15, respectively. The green shadings represent the span of the reported experimental TOF. Inset: One example of a strain configuration for each $\sigma$ value. Tensile strain is marked in red, whereas compressive strain is marked in blue. Deeper color represents a higher strain. Temperature: 673 K, Pressure: $p_{N_2} = 2.5$ bar, $p_{H_2} = 7.5$ bar.

significantly affect the TOF on Ru(0001), whereas the impact on the stepped surface is smaller. Thus, the TOF difference between the Ru(0001) and the stepped surface is attributed to overestimating the TOF of the Ru(0001) surface. Sites with properties similar to the Ru(0001)-sites are not abundant on Ru NP, which limits the effect of the overestimated TOF over Ru(0001) when using the linear scaling relations. (A comparison between the TOFs evaluated with scaling relations and explicit DFT-calculations for a Ru₇₈₀ NP is given in Supplementary Table 25).

The TOF over unstrained Ru₇₈₀ is calculated to be $3.3 \times 10^{-4}$ site⁻¹ s⁻¹, which is about one order of magnitude higher than for the stepped surface. The higher catalytic activity compared to the stepped surface is a consequence of the wide distribution of site types and kinetic couplings on Ru NPs. The TOF on the unstrained Ru₇₈₀ is still, however,

about one order of magnitude lower than the experimental values. One difference between the simulated and experimental TOF is the absence of inherent strain in the simulated NP. To quantitatively evaluate the effect of strain on the TOF, the influence of strain on the reaction energy is described by a linear scaling relation[34]. Here, the changes in reaction energies and energy barriers caused by strain are added to each elementary reaction. The parameters in the linear scaling relations are fitted by DFT calculations on Ru(0001) and a stepped Ru surface (see Supplementary Table 18) and confirmed for an NP (see Supplementary Fig. 9). For the $N_2$ dissociation step, applying tensile (compressive) strain results in lower (higher) energy barriers. On the contrary, for the $NH_x$ hydrogenation steps, tensile (compressive) strain leads to higher (lower) energy barriers. The opposite trends of energy barriers with strain on $N_2$ dissociation and $NH_x$ hydrogenation steps suggest that strain could influence elementary reactions in different ways. To simulate the kinetics over Ru NPs with a distribution of strained atoms, a random normal distribution is applied:

$$f(\Delta s) = \frac{1}{\sigma\sqrt{2\pi}} \exp\left[-\frac{1}{2}\left(\frac{\Delta s - \mu}{\sigma}\right)^2\right] \quad (8)$$

Where $\Delta s$ is the strain, which is defined by the nearest neighbor difference with respect to bulk, $\mu$ is the mean value of strain, and $\sigma$ is the standard deviation.

Here, we first set the value of $\mu$ to zero and change the width of strain distributions by modifying $\sigma$. With a $\sigma$ value of 0.05, the TOF is calculated to be $3.2 \times 10^{-3}$ site$^{-1}$ s$^{-1}$. Increasing $\sigma$, results in a further increase of the TOF. The incorporation of randomly distributed strain in Ru$_{780}$ makes the simulated TOF fall within the experimental range, highlighting the importance of inherent strain in kinetic simulations. Besides the TOF, the apparent activation energy is another kinetic parameter when evaluating kinetic models. Here, an Arrhenius analysis is performed in the temperature range of 623 K to 723 K (see Supplementary Fig. 2). The apparent activation energies obtained on unstrained and randomly ($\sigma = 0.05$) strained Ru$_{780}$ are 0.73 eV and 0.69 eV, respectively, which both are within the experimental range of 0.6 eV–1.0 eV (see Supplementary Table 10). The calculated TOF and apparent activation energy are within the range of experimental reports, which indicates that the kinetic model properly describes the experimental situation and consolidates the importance of strain effects.

In the randomly strained NP, different strain configurations lead to different TOFs. To investigate the importance of the detailed strain configuration, Fig. 2b shows the TOF distribution on 256 randomly strained Ru NPs at $\sigma$ values of 0.05, 0.10, and 0.15. The TOF distribution of the randomly strained Ru NPs follows a normal distribution. The distributions for the three $\sigma$ values are well separated, with only a slight overlap between $\sigma = 0.10$ and 0.15. Nearly all TOF values in the distributions fall within the experimental range. Comparing the distributions at different $\sigma$ values, the average TOFs at $\sigma$ of 0.05, 0.10 and 0.15 are calculated to be $3.6 \times 10^{-3}$ site$^{-1}$ s$^{-1}$, $1.2 \times 10^{-2}$ site$^{-1}$ s$^{-1}$ and $1.9 \times 10^{-2}$ site$^{-1}$ s$^{-1}$, with the standard deviation of 0.00079, 0.0016 and 0.0021, respectively. A wide strain distribution results in a higher TOF, which suggests that broadening the strain distribution is one approach to tailor high-activity Ru catalysts.

In addition to random strain, particles may also be homogeneously strained. To evaluate the impact of homogeneous strain by modifying $\mu$, we first concentrate on the case with Ru$_{780}$ without random strain (blue line in Fig. 2a). The TOF exhibits a volcano-shaped dependence on the strain with a maximum ($1.8 \times 10^{-3}$ site$^{-1}$ s$^{-1}$) located at a tensile strain of 4%. The TOF is at the maximum enhancement by a factor of 5 with respect to the unstrained Ru NPs. To explore the effect of both random and homogeneous strain, we apply the random strain distribution to every homogeneous strained NP with the $\sigma$ value set to be 0.05 (red line in Fig. 2a). The introduction of random strain further

increases the TOF, however, the maximum is still at 4% tensile strain being $3.8 \times 10^{-3}$ site$^{-1}$ s$^{-1}$. The Ru NP with both random and homogeneous strain has a TOF that is 12 times that of the unstrained NP. As compared to unstrained particles, it is possible, with strain, to lower the temperature while maintaining the TOF, which is important from an energy-saving perspective. The randomly and homogeneously strained Ru NP reaches the same TOF at 583 K as the unstrained Ru NP at 673 K.

## Reaction mechanisms

Knowing that strain significantly changes the kinetics of $NH_3$ synthesis over Ru NP it becomes important to analyze the reaction mechanisms in detail. We start by comparing the normalized rate for $N_2$ dissociation and $NH_2$ hydrogenation of unstrained Ru$_{780}$ as a function of GCN, Fig. 3a. $N_2$ dissociates predominantly on high GCN sites owing to the low energy barrier. $NH_2$ hydrogenation occurs mainly on low GCN sites because of the stronger binding of $NH_x$ and H species on low GCN sites. $NH_x$ species diffuse from high to low GCN sites during the reaction, which stresses the crucial role of kinetic coupling between different regions on NPs. The normalized rates for the two elementary reactions for homogeneous strained and randomly strained Ru NPs exhibit similar trends as the unstrained NP (see Supplementary Fig. 3).

To analyze which elementary reaction step controls the overall reaction and how this is affected by strain, we perform a degree of rate control (DRC) analysis[35]. Because the DRCs do not add up to one, which is a known issue in kMC simulations[36], we are focusing on trends. To facilitate the analysis, we explore simplified models of the NPs in the form of surface models that contain only two types of sites, with GCN 4.0 and 8.0 (Fig. 3b). We have verified that the simplified models capture the trend of TOF with respect to strain (see Supplementary Fig. 4). $N_2$ dissociation and $NH_2$ hydrogenation are the two elementary reactions with dominating DRCs. For the unstrained surface model, $N_2$ dissociation has a DRC of 0.85, whereas the DRC of $NH_2$ hydrogenation is 0.08. The situation is changing for the model systems with homogeneous tensile strain, in which the $N_2$ dissociation is not the sole rate controlling step. On the NP with 8% tensile homogeneous strain, the combined DRC for $NH_2$ hydrogenation on sites with GCN of 4.0 and 8.0 are higher than the DRC of $N_2$ dissociation. Thus, our analysis shows that strain changes the rate-controlling step for $NH_3$ synthesis over Ru NPs.

In a system with only one type of site, every elementary reaction must occur on one common potential energy surface, which means that the reaction activity can be described by a volcano curve. The result that strain changes the rate controlling step for $NH_3$ synthesis over Ru NPs can be understood by a schematic volcano analysis, Fig. 3c. The right and left sides of the volcano are controlled by $N_2$ dissociation and $NH_2$ hydrogenation, respectively. Applying tensile strain, i.e., moving to the left in the volcano, increases the TOF for $N_2$ dissociation, whereas the TOF for $NH_2$ hydrogenation is decreased. On unstrained Ru NPs, $N_2$ dissociation occurs mainly on high GCN sites (blue solid line), whereas $NH_2$ hydrogenation occurs on low GCN sites (orange solid line). Homogeneous tensile strain promotes the reaction rate for $N_2$ dissociation, whereas $NH_2$ hydrogenation is slowed down. Having a system with site communication, the maximum TOF is obtained when the $N_2$ dissociation and $NH_2$ hydrogenation have the same reaction rate (blue and orange dashed lines). For an NP, the maximum can be reached by applying a certain value of tensile strain. Tensile strain beyond this value will decrease the TOF because $NH_2$ hydrogenation becomes the rate-limiting step.

Inhomogeneously strained Ru$_{780}$ can be used to further analyze how different sites contribute to the TOF, Fig. 3d. Adding compressive strain to low GCN sites, which promotes the $NH_x$ hydrogenation steps, does not affect the TOF. The reason for the unchanged TOF is that $N_2$ dissociation controls the reaction for an otherwise unstrained NP. Instead, applying tensile strain to high GCN sites enhances the TOF

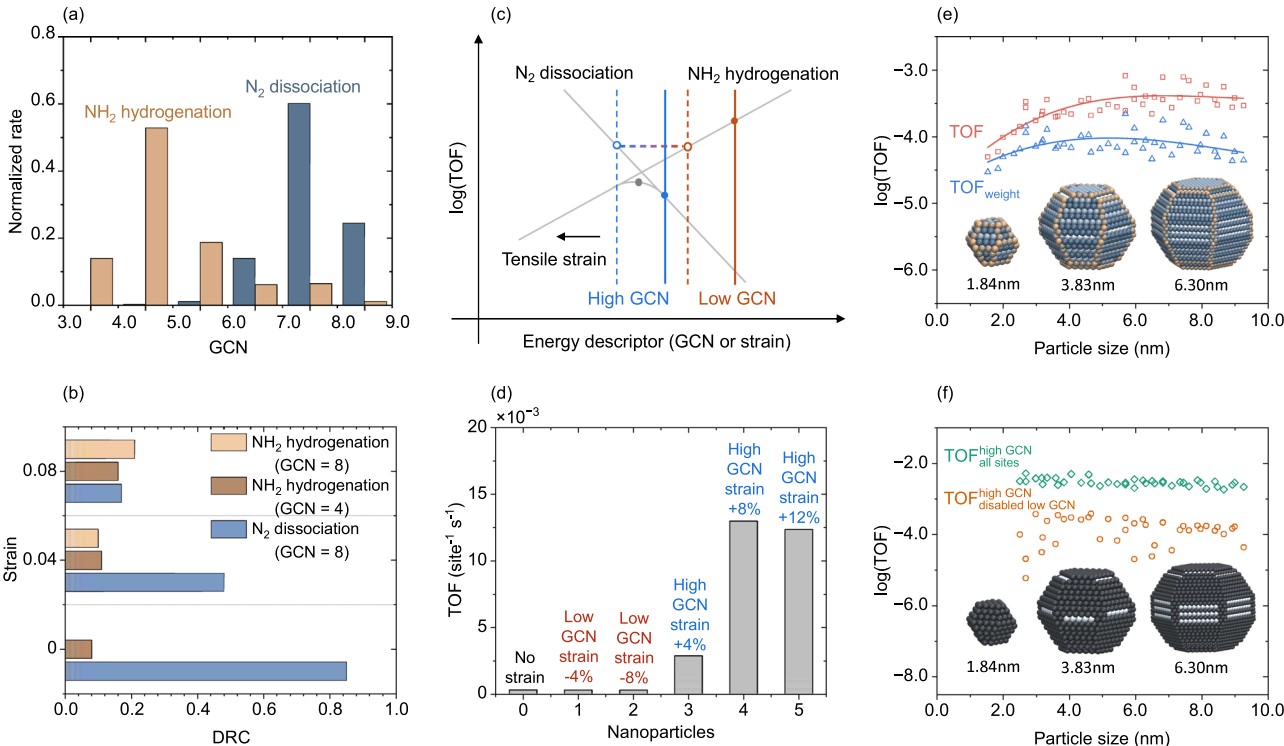

**Fig. 3 | Analysis of NH₃ synthesis reaction. a** Normalized rates of N₂ dissociation and NH₂ hydrogenation steps versus GCN. **b** Degree of Rate Control (DRC) analysis versus strain on simplified models with GCN 4.0 and 8.0. The contribution to the DRC from N₂ dissociation on GCN = 4 sites is negligible and have been omitted. The NH₂ hydrogenation on GCN = 8 is small and not visible. **c** Schematic volcano analysis on Ru NPs. **d** TOF on inhomogeneous strained Ru NPs. **e** TOF versus particle sizes for NPs with respect to the number of surface atoms (red squares) and the number of particle atoms (blue triangles). The curves are fitted by cubic polynomials. **f** TOF versus particle sizes with respect to high GCN sites. The TOFs are calculated by keeping all sites active (green diamonds) and disabling low GCN sites (orange circles). Temperature: 673 K, Pressure: $p_{N_2}$ = 2.5 bar, $p_{H_2}$ = 7.5 bar.

significantly. However, a tensile strain on high GCN sites exceeding 8% does not further increase the TOF, which is a consequence of NH₂ hydrogenation becoming rate-controlling. In this case, applying compressive strain on low GCN sites can notably promote the TOF (see Supplementary Fig. 5). For randomly strained Ru NPs, the co-existence of tensile and compressive strains on different sites on the NP simultaneously promote the N₂ dissociation and NH₂ hydrogenation steps, which overall enhances the catalytic activity of the Ru NPs.

An NP has a collection of different types of sites (GCNs) where the ratio between edge and facet sites changes with particle size. The size-dependent TOF for Wulff-shaped Ru NPs in the size range of 1.53 nm–9.26 nm are shown in Fig. 3e. The TOF with respect to the number of surface atoms (denoted TOF in the figure) increases with particle size up to 3 nm, after which it slightly decreases. The decrease in TOF after 3 nm is consistent with previous experiments[37]. The lower TOF for particles below ~2.5 nm is a consequence of the low abundance of GCN > 8 on small particles. Once the particle size exceeds 3 nm, the high GCN sites appear on both edge sites and Ru(10$\bar{1}$0) facet sites, which promotes the reaction activity. As Ru is a precious metal, it is important to optimize the NP size with respect to the total number of Ru atoms in the NP. The TOF with respect to the total number of particle atoms (denoted TOF$_{weight}$) shows a maximum at 5 nm. Thus, considering the scarcity of Ru metals, Ru NP should be synthesized to have a particle size of 5 nm for the ammonia synthesis reaction.

The result that N₂ dissociation and NH₂ hydrogenation preferably occur on different sites (Fig. 3a, b) stresses the importance of site communication and kinetic couplings. To further elucidate the roles of different sites on the NPs, we compare the TOF of the NPs with the case where low GCN sites (GCN < 8) artificially has been disabled, Fig. 3f. The TOF is, in this case, calculated with respect to the high GCN sites (GCN > 8). The TOF with all sites active (TOF$_{all\ sites}^{high\ GCN}$) does not have any

size dependence, which underlines the importance of GCN > 8 for N₂ dissociation. Importantly, however, the TOF is reduced by more than one order of magnitude if the low GCN sites are disabled (TOF$_{disabled\ low\ GCN}^{high\ GCN}$). By investigating the detailed reaction mechanism on the NP with disabled low GCN sites, we find that the NH₂ hydrogenation is rate limiting. Thus, the presence of low GCN sites on the NPs serves as sites for NH₂ hydrogenation and is important for the overall TOF. The result clearly shows that different sites are important for the NH₃ synthesis reaction, making it crucial to consider the assembly of sites when analyzing and simulating NH₃ synthesis over Ru NP. It is the assembly of sites rather than one particular site that governs the activity.

### Effect of strain on NPs with different sizes and shapes
Technical catalysts contain NPs with a distribution of sizes and shapes. To understand how the effects of strain depend on particle size, we simulate the TOF for five homogeneously strained Wulff-shaped particles, Fig. 4a. The five sizes have similar strain dependencies with a maximum TOF located at a tensile strain of + 4%. The result shows that the influence of strain is independent of the particle sizes. In agreement with Fig. 3e, the TOF has a weak particle size dependence as long as sites active for N₂ dissociation are present. To explore the effects of shape, five different types of NPs with diameters of about 2.68 nm: Wulff-shaped, octahedron, icosahedron, decahedron, and cube (Fig. 4b). The different shapes have a maximum in the TOF at about + 4% tensile strain, showing that the effect of strain is independent of the shapes of NPs. The ordering in TOF between the shapes is preserved for the different stains, with the exception of high tensile strains where the cube becomes slightly more active than the decahedron. We note that the Wulff-shaped NPs exhibit the highest TOF for all strains. The superior performance of the Wulff-shaped NPs is related to the

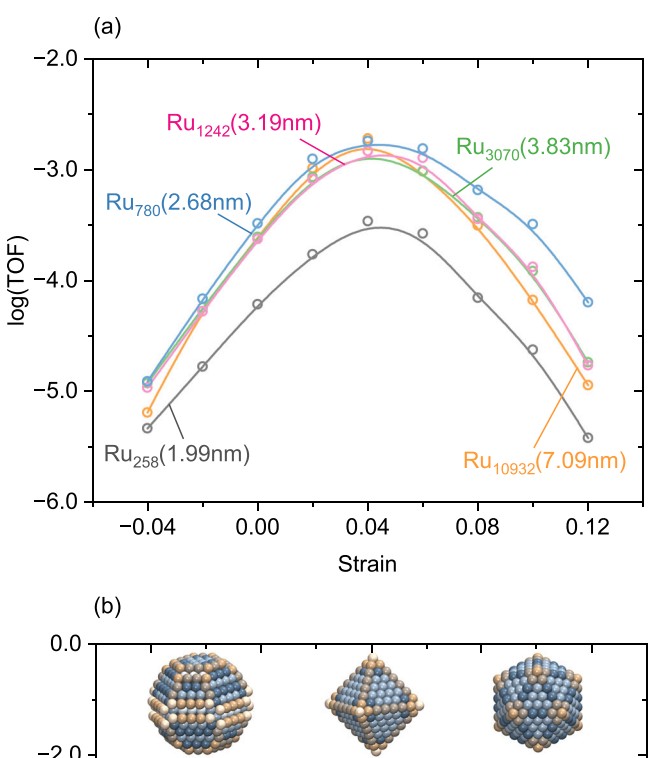

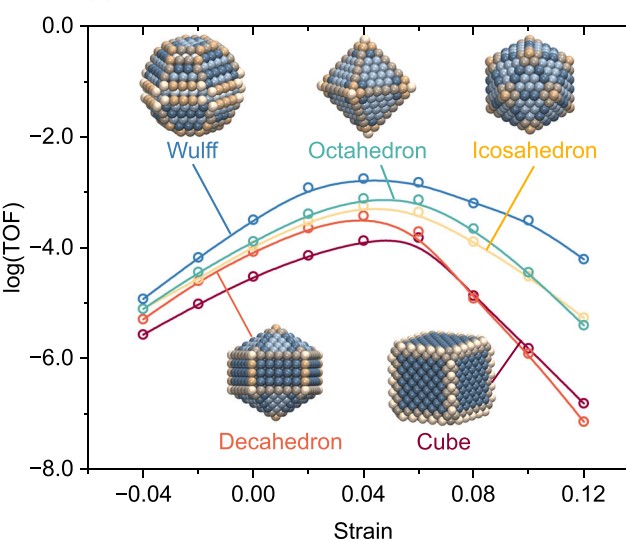

**Fig. 4 | Analysis of how homogeneous strain affect NH₃ synthesis.** TOF versus strain of Ru NPs with different (**a**) sizes and (**b**) shapes. Temperature: 673 K, Pressure: $p_{N_2}$ = 2.5 bar, $p_{H_2}$ = 7.5 bar.

makes it possible to propose new strategies for designing metal-efficient Ru catalysts. Ru NPs should be supported to have a high degree of inherent strain, which potentially could be fabricated on supports with a strong metal-support interaction.

## Methods

The KMC simulations are carried out using the MonteCoffee program[38] based on coarse-grained adsorption sites. Each coarse-grained entity contains the hollow, bridge, and top positions, and the surface species are considered in the preferred geometric adsorption site. For the extended Ru(0001) surface and a surface with steps, the KMC simulations are carried out using 16 × 10 and 17 × 10 lattice models, respectively, with periodic boundary conditions (see Supplementary Fig. 6). The Wulff construction Ru NPs in KMC simulations are built via the WulffPack code[39].

The energy parameters are obtained by DFT calculations with the BEEF-vdW exchange-correlation functional[40]. All DFT calculations are performed by using the Vienna Ab initio Simulation Package (VASP)[41–44] using the projector-augmented wave (PAW) scheme[45], and the electronic states were expanded on a plane wave basis with a cutoff of 400 eV. To model the Ru surfaces and NPs, the electronic energies were calculated on Ru(0001), Ru(10$\bar{1}$0), Ru(10$\bar{1}$1), Ru(10$\bar{1}$2) and Ru(20$\bar{2}$1). In addition, the $A_5$, $B_{5,}$ and $D_5$ steps are constructed from Ru(0001). Ru(0001) is also used to construct an edge. The reference energies of gaseous N₂ and H₂ are optimized using a (10 × 10 × 10) Å box. All the reaction energies used in kinetic simulations are zero-point energy corrected. Vibrational analysis is performed within the harmonic approximation using finite differences.

In the kMC simulations, the reaction barriers for unstrained Ru NPs are obtained using the scaling relation in Equation (9)[46,47]. In this way, the reaction barriers ($E_a$) are given as a function of reaction energy ($\Delta E$), which is described by the linear scaling relations using GCNs[48,49], in Equation (10).

$$E_a = a\Delta E + b \tag{9}$$

$$\Delta E = c \cdot GCN + d \tag{10}$$

Where a, b, c, and d are the parameters fitted by the DFT calculations. The GCN is defined as:

$$GCN_i = \frac{1}{cn_{max}} \sum_{j=1}^{n_i} cn_j \tag{11}$$

Where $n_i$ is the number of nearest neighbors of site $i$. $cn_j$ is the coordination number of neighboring site $j$, and $cn_{max}$ is the maximum coordination number for atoms in hcp Ru bulk. The fitting parameters for the linear scaling relations are given in Supplementary Tables 11 and 12, respectively. The kinetics results have a weak dependence on the slopes of the scaling relations, as demonstrated in Supplementary Tables 23.

The adsorption processes of gaseous N₂ or H₂ are considered to be barrierless with rate constants described by collision theory:

$$k^{ads} = \frac{pA_{site}}{\sqrt{2\pi m k_B T}} \tag{12}$$

Where $p$ is the partial gas pressure, $A_{site}$ = 6.15 Å² is the area of the adsorption site. To ensure the thermodynamic consistency for reversible reactions, the rate constants for the desorption process are calculated by the equilibrium constant:

$$K^{ads} = \frac{k^{ads}}{k^{des}} = \exp\left(-\frac{\Delta G^{ads}}{k_B T}\right) \approx exp\left(-\frac{\Delta E^{ads} - T\Delta S^{ads}}{k_B T}\right) \tag{13}$$

wide distribution of site types, which provide active sites for both N₂ dissociation and the NHₓ hydrogenation steps. Considering that the Wulff-shaped NPs combine high activity and stability, it is the preferred shape for Ru NP in ammonia synthesis.

We have performed first-principles-based kMC simulations to study ammonia synthesis over Ru NPs. By accounting for the combined effect of structural complexity and inherent strain, we can obtain TOFs consistent with experimental data. Kinetic couplings, where sites with different properties communicate by adsorbate diffusion, are found to be crucial. N₂ dissociation occurs mainly on highly coordinated sites, whereas NHₓ hydrogenation occurs preferably on undercoordinated sites. We find a maximum in the TOF for a homogeneous tensile strain of 4% for different particle shapes and sizes. The effect of strain is not sensitive to particle shape and size, which suggests that the conclusions are general for the ammonia synthesis reaction. Technical catalyst particles are often measured to have an inherent random strain distribution, and we find that it is crucial to account for this distribution when evaluating the TOF. The detailed understanding of how kinetic couplings and strain affect the ammonia synthesis reaction

Where $\Delta G^{ads}$ is the Gibbs free energies, $\Delta S^{ads}$ is the adsorption entropy, and $\Delta E^{ads}$ is the adsorption energy of the adsorption process.

The rate constants for the surface reactions are obtained by harmonic transition state theory:

$$k = \kappa(T) \frac{k_B T}{h} exp\left(-\frac{\Delta G^{act}}{k_B T}\right) \approx \kappa(T) \frac{k_B T}{h} exp\left(-\frac{\Delta E^{act} - T\Delta S^{act}}{k_B T}\right)$$

(14)

Where $\Delta G^{act}$ is the Gibbs free energy barrier, $\Delta S^{act}$ is the entropy barrier, $\Delta E^{act}$ is the energy barrier, and $h$ is Planck's constant. $\kappa(T)$ is 1 for the $NH_x$ hydrogenation reactions. For the $N_2$ dissociation step, $\kappa(T)$ is fitted to experiments[12,15] to account for the low measured sticking probability. Further details on entropy estimations, energy parameters with strain effect, lateral interactions, the scaling factor $\kappa(T)$, and DRC analysis are described in Supplementary Methods, respectively.

The barriers for adsorbate diffusion are known to be a fraction (about 0.12) of the adsorption energy[50]. The diffusion rates are for all considered adsorbates considerably faster than the rates for $N_2$ dissociation and $NH_2$ hydrogenation. The diffusion rates has, therefore, been scaled-down, while maintaining the quasi-equilibrium.

## Data availability

The structural data generated in this study have been deposited in the Zenodo database with accession code[51] https://doi.org/10.5281/zenodo.14670930.

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

## Acknowledgements

Financial support is acknowledged by the Swedish Research Council (2020-05191) and Chalmers Area of Advance Nano. The calculations were performed at NSC via a NAISS grant (2022/3-14). The Competence Center for Catalysis (KCK) is hosted by Chalmers University of Technology and is financially supported by the Swedish Energy Agency and the member companies Johnson Matthey, Perstorp, Powercell, Preem, Scania CV, Umicore, and Volvo Group.

## Author contributions

All the authors conceived the study. Y.Y. performed the calculations and simulations and drafted the manuscript. H.G. and A.H. supervised the work and edited the manuscript.

## Funding

## Competing interests

The authors declare no competing interests.
