## [Transparent Peer Review File · Nature Communications]

Inherent strain and kinetic coupling determine the kinetics of ammonia synthesis over Ru nanoparticles

Corresponding Author: Professor Henrik Gronbeck

Version 0:

Reviewer comments:

Reviewer #1

(Remarks to the Author)

This manuscript studied catalytic activity of Ru nanoparticles towards ammonia synthesis by DFT-kMC methods. Authors discussed the effects of strain distribution and coordination number on the activation barrier, reaction energy, and reaction rate of ammonia synthesis. Authors claim that the different elementary reaction steps take place on different active sites, and the strain can cause contrary trends for N₂ dissociation and N hydrogenation.

1, how to define the adsorption energy? Why NH₃ has higher adsorption energies than NH, since NH is unsaturated N-H adsorbate? For GCN is larger than 8.0, no data is provided in Fig.1a for NH₂ and NH₃'s scaling relations. How about the uncertainty of scaling relations to predict the adsorption energies and activation barriers (in Fig.1a and Fig.S1)? Why the parameter of BEP is negative for N₂ dissociation in Fig.1a? what's the physical mechanism?

2, since the "The potential energy surfaces derived from linear scaling relations underestimate the difference in the N₂ dissociation barrier over Ru(0001) and a stepped Ru surface", the potential energy surfaces in Fig.1b are not reliable. Moreover, the uncertainty of scaling relations inevitably cause larger errors for predicting the catalytic activity of ammonia synthesis reaction, especially under strain loading.

3, the linear scaling relations for reaction energies and activation barriers with respect to strain are obtained on Ru extended surfaces. Considering the different atomic coordination numbers on Ru NP, the reliability of these relations is questionable to predict the reaction energies and activation barriers.

4, how to define the strain on NP? especially for homogeneous strain? Due to the different atomic coordination numbers on Ru NP, the bond lengths are different on NP.

Therefore, I don't recommend this manuscript to be published with current version.

Reviewer #2

(Remarks to the Author)

The manuscript reports a theoretical study of ammonia synthesis on Ru nanoparticles. The study reveals a few important theoretical observations, especially the strain-TOF relation. The strain-TOF relation is due to kinetic couplings between strain and catalytic site contribution. The strain-TOF relation is independent of particle size. In the study, the authors mainly use kinetic Monte Carlo simulations based on potential energy surfaces fitted to density functional theory calculations. I agree that these theoretical insights offer strong support for the design of new catalysts. Given that ammonia synthesis is such an important and widely concerned topic, I believe the study is timely and suitable for the journal. The manuscript is well written. However, before I can recommend acceptance, the following points should be addressed to ensure that the theoretical methods and analyses are sound.

1. An important assumption applied in the study is the linear relations between the defined GCN and reaction energy. Can

the authors justify why such linear relations should apply? As observed in Fig. 1a, there are outliers. The authors have discussed the outlier indicated by the solid marker, and have claimed (line 151) that it does not affect the kinetics simulation because the corresponding surface is not abundant. I have a few more questions related to this. Can the authors provide a more quantitative assessment of the impact of the outlier on the final results? Can the authors provide some insight into why the linear relation breaks significantly for this particular point? Can the authors check whether the discrepancy has something to do with the BEEF-vdW DFT functional used (or other settings, which are less likely)?

2. The authors use a random strain model, which allows them to obtain TOF in the regime of experimental results. Can the authors justify why a random strain model is physically reasonable? My intuition is that the strain field in nanoparticles should be homogeneous or some continuously distributed field.

3. The authors claimed agreement with experiment, but based on my reading the agreement is only that the calculated TOF is in the same order of magnitude as experimental measurements. If I did not miss other information, then I would suggest the authors make the statements clearer. Given the agreement comes along with a few assumptions employed (linear relations and random strain model), one should be careful not to over-interpret the importance of “theory-experiment” agreement.

4. In Fig. 1a and Fig. 2a, the data points are not clearly indicated in the caption, they are mentioned in the main text and SI table, I suggest the authors add more description sentences in the captions to improve the readability of the figures alone, where they can also refer to corresponding information in the SI explicitly.

5. Figure 1b depicts the potential energy surfaces of sites with varying GCN. Is it possible, from either an energetic or KMC perspective, for two NH_3 molecules to form simultaneously (e.g., $\text{NH}_2^* + \text{NH}^* + 3\text{H}^*$)? It would benefit the authors to explicitly state whether these potential energy surfaces represent a reasonable simplification or if they merely illustrate one of several possible reaction pathways.

6. The statement in Line 141, “The difference in TOF between Ru(0001) and the stepped surface is only 20” suggests a difference in absolute value, which should be clarified.

7. Considering strain in simulations is a significant contribution. The authors also employ linear relations to consider the strain effects on reaction energy and energy barrier. The linear parameters are fitted from DFT calculations, but the data for fit should be shown.

8. Additionally, can the authors comment on the statistical error when they use a normal distribution to model randomly strained atoms.

9. In Figure 3(b), are the columns for N_2 dissociation (GCN=4) and NH_2 hydrogenation (GCN=8, strain=0) omitted due to their negligible values?

10. Planck constant h in equation 6 and 7 is not mentioned.

11. It is not clear what the authors mean by “break the limitation of linear scaling relations” in the abstract and conclusions.

Reviewer #3

(Remarks to the Author)
Please see attached.

Version 1:

Reviewer comments:

Reviewer #2

(Remarks to the Author)
The authors have addressed my comments, and I believe that with the revisions and additions the manuscript can be accepted for publication at Nature Communications.

Reviewer #3

(Remarks to the Author)
The authors have revised the manuscript satisfactorily, publication is recommended.

We thank the Reviewers for their valuable comments, which have helped us to improve the manuscript. A point-by-point response to all comments is included below. The Reviewers' comments are in blue, our responses are in black, and the changes made to the manuscript are indicated in red.

Response to Reviewer 1:

This manuscript studied catalytic activity of Ru nanoparticles towards ammonia synthesis by DFT-kMC methods. Authors discussed the effects of strain distribution and coordination number on the activation barrier, reaction energy, and reaction rate of ammonia synthesis. Authors claim that the different elementary reaction steps take place on different active sites, and the strain can cause contrary trends for N₂ dissociation and N hydrogenation.

1, how to define the adsorption energy? Why NH₃ has higher adsorption energies than NH, since NH is unsaturated N-H adsorbate? For GCN is larger than 8.0, no data is provided in Fig. 1a for NH₂ and NH₃'s scaling relations. How about the uncertainty of scaling relations to predict the adsorption energies and activation barriers (in Fig. 1a and Fig.S1)? Why the parameter of BEP is negative for N₂ dissociation in Fig. 1a? what's the physical mechanism?

Response 1: We thank the Reviewer for the questions regarding the evaluation of the potential energy surface.

The adsorption energy of every surface species is defined by the relative energy with respect to gaseous N₂ and H₂. The adsorption energy of NH_x (ΔE_{NH_x} , x = 0 - 3) species is, thus, calculated according to:

$$\Delta E_{NH_x} = E_{NH_x-surface} - E_{surface} - \frac{1}{2}E_{N_2(g)} - \frac{x}{2}E_{H_2(g)} \quad (1)$$

Where $E_{NH_x-surface}$ is the total energy of the Ru surface with NH_x surface species, $E_{surface}$ is the energy of the clean surface, $E_{N_2(g)}$ and $E_{H_2(g)}$ are the energies of N₂ and H₂ in the gas phase, respectively. We have added the definition of adsorption energy in the manuscript on page 4:

Here, the potential energy surface is described by scaling relations using the site-specific generalized coordination number (GCN), see Figure 1a. The linear relations are fitted to DFT calculations considering each surface species in the stable adsorption sites on different surfaces, with respect to gaseous N₂ and H₂, see Supplementary Information (Supplementary Tables 1-6).

Moreover, the Supplementary Information has been modified to include the equation for the evaluation of the adsorption energies.

The adsorption energy depends on the references, which in our case are gaseous N₂ and H₂ and the clean Ru surfaces. Thus, the adsorption energy reflects both the bond strength between the adsorbate and the surface and the stability of the NH_x species with respect to gaseous N₂ and H₂. If the adsorption energy of NH_x instead were defined with respect to gaseous NH_x species, the adsorption energies of NH would be more negative than that of NH₃. It is clear that the choice of references does not affect the kinetic results. Our choice is convenient and standard within computational catalysis as it does not include the evaluation of energies for radicals.

We calculate the site-specific GCN of surface species in their stable adsorption configurations on Ru(0001), Ru(10 $\bar{1}$ 0), Ru(10 $\bar{1}$ 1), Ru(10 $\bar{1}$ 2), Ru(2 $\bar{1}$ $\bar{1}$ 2), as well as the A₅, B₅, and D₅ steps together with an edge-model. The stable adsorption configurations of NH₂ and NH₃ are bridge and atop sites, respectively. The GCNs of these configurations are smaller than 8.0. Thus, adsorption configurations with GCN higher than 8.0 are not stable. In the kMC calculations, we use NPs with GCNs ranging from 3.16 to 9.16. Thus, the linear scaling relations for NH₂ and NH₃ are extrapolated for GCN values above 8.0. However, as adsorption configurations with GCN higher than 8.0 are energetically unfavorable, such sites will rarely be occupied by any of the reaction intermediates. Thus, the extrapolated values will not affect the reaction kinetics. We have in the revised manuscript clarified from which data the scaling relations were obtained (page 19):

To model the Ru surfaces and NPs, the electronic energies were calculated on Ru(0001), Ru(10 $\bar{1}$ 0), Ru(10 $\bar{1}$ 1), Ru(10 $\bar{1}$ 2), Ru(2 $\bar{1}$ $\bar{1}$ 2). In addition, the A₅, B₅ and D₅ are constructed from Ru(0001). Ru(0001) is also used to construct an edge.

We thank the reviewer for the comment on the uncertainty in the scaling relations. This question has motivated us to perform a thorough analysis of how the results are affected by the slopes in the scaling relations. To assess the uncertainty in adsorption energies and activation barriers in Figure 1a and Supplementary Figure 1, we performed a sensitivity analysis on the slope of the scaling relations of N atom, NH, NH₂ and H atom on the Ru₇₈₀ NP. As the activation barriers are computed relative to the reaction energies (with scaling relations), the sensitivity analysis on linear scaling relations also captures the uncertainty in activation barriers. In this analysis, we first changed the slope of the scaling relations for N atom, NH, NH₂ and H atom by $\pm 10\%$ and $\pm 20\%$, while keeping other energy parameters constant. The different cases are shown in Figure R1.

Figure R1: Linear scaling relations with different slopes.

The results of sensitivity analysis on the TOF of Ru₇₈₀ NP are shown in Table R1. The relative TOF difference (ΔTOF) is calculated according to Equation 2,

$$\Delta TOF = \frac{TOF - TOF_{unchanged}}{TOF_{unchanged}} \times 100\% \quad (2)$$

Where $TOF_{unchanged}$ is the TOF calculated with the original slope.

For NH and NH₂, the changes in slopes have negligible effect on the kinetics, with the TOF differences falling within the standard deviation of kMC simulations (about 3%). For the H atom, adjusting the slope by 20% leads to a TOF difference of about 15%. The effect of the H adsorption energy on ΔTOF is understandable, as it affects the availability of free sites. For the adsorption energy of N, a change of the slope by 20% yields a ΔTOF of about 25%, as the N adsorption energy affects the activation barrier of N₂ dissociation through the linear scaling relations. Thus, the change in ΔTOF is a natural consequence of N₂ being the rate-controlling step under most conditions.

The sensitivity analysis shows that the slope variations in the linear scaling relations on each surface species have small effects on the TOF. The small effects can be attributed to the wide distribution of sites (many different GCNs) on Ru NPs. For example, an increase in the slope of N adsorption energy raises the N₂ dissociation activation barrier at high GCN sites

but reduces it at low GCN sites. This site-specific compensation effect reduces the overall influence of energy changes on the reaction kinetics.

Table R1. Relative TOF difference (ΔTOF) by each surface species.

Surface species	Slope changes				
	-20%	-10%	0	+10%	+20%
NH	4.82%	2.58%	0	4.00%	1.43%
NH ₂	0.47%	3.58%	0	5.05%	5.23%
H	15.33%	7.33%	0	1.92%	-3.91%
N	26.19%	14.65%	0	-8.87%	-16.43%

It is well known [Phys. Rev. Lett. 99, 016105 (2007)] that the linear scaling relations for the adsorption energies of N, NH, NH₂ and NH₃ are correlated. To investigate the combined effects of NH_x surface species, we simultaneously adjusted the slopes in the scaling relations by $\pm 10\%$ and $\pm 20\%$. The kinetic results by kMC simulations are shown in Table R2. As the slopes for NH, NH₂ and NH₃ have negligible impact on the kinetics, the overall effect is similar to that of changing the slope of the N atom.

Table R2. Relative TOF difference (ΔTOF) adjusting the scaling relations for the adsorption energies simultaneously for N, NH, NH₂ and NH₃.

Slope changes				
-20%	-10%	0	+10%	+20%
16.98%	8.67%	0	-9.65%	-13.46%

Our sensitivity analysis shows that the TOF is not sensitive to the slope of linear scaling relations, suggesting that the uncertainties in linear scaling relations in Figure 1a and Supplementary Figure 1 have minor influence on the kinetic results for the NPs. The sensitivity analysis is added in the revised version of the Supplementary Information. Moreover, we have added a sentence in the main text that a sensitivity analysis has been done (page 20):

The kinetics results have a weak dependence on the slopes of the scaling relations, as demonstrated in the Supplementary Table 23.

It is true that the calculated negative slope for the N₂ activation energy with respect to the reaction energy is unusual. However, BEP relations are generally reported for one type of site for different metals [J. Catal. 197, 229-231 (2001)]. Here, we are investigating different types of sites for one metal. For N₂ dissociation, the physisorbed N₂ molecule (initial state) has a small influence on the scaling relations. The dissociation barrier depends instead on the geometry of the transition state and, in particular, how well the Ru surface matches the transition state geometry. Different types of Ru surfaces provide structures that match the

transition state differently. Thus, the N₂ dissociation barrier does not fully follow the definition of BEP relations, which relates the activation energy with the reaction energy.

The negative slope of the linear scaling relations for N₂ dissociation step can be understood by considering the coordination number of the sites for dissociation. Generally, the high-coordination-number sites have lower adsorption energy. This applies also to the adsorption energy of N atoms (final state). However, for the transition state of N₂ dissociation, the high-coordination-number cavity atoms offer a stabilization of the transition state. As a result, a lower activation barrier corresponds to a lower reaction energy in this linear scaling relation. This behavior is different from the BEP relations fitted across different metals but with similar transition state structures.

To avoid misunderstandings, we have modified the manuscript to distinguish scaling relations from BEP relations.

On page 5:

“Linear scaling relations are used to evaluate the transition state energies as a function of the reaction energies. The scaling relation for N₂ dissociation has a slope of -0.38, which means that a high reaction energy corresponds to a low energy barrier (inset Figure 1a).”

In Page 19:

“In the kMC simulations, the reaction barriers for unstrained Ru NPs are obtained using the scaling relation in Equation 2.”

In Page 20:

“The fitting parameters for the linear scaling relations are given in Supplementary Tables 11 and 12, respectively.”

2, since the “The potential energy surfaces derived from linear scaling relations underestimate the difference in the N₂ dissociation barrier over Ru(0001) and a stepped Ru surface”, the potential energy surfaces in Fig.1b are not reliable. Moreover, the uncertainty of scaling relations inevitably cause larger errors for predicting the catalytic activity of ammonia synthesis reaction, especially under strain loading.

Response 2: We thank the Reviewer for the comment about the result for N₂ dissociation over Ru(0001). As shown above, the kinetic results are not sensitive to the slopes of the scaling relations. Regarding the Ru(0001) sites, we note that these sites (GCN=7.5) are minority sites on the Ru NPs, with a low contribution to the overall TOF. To make this clearer, we have revised the text on page 5 in the manuscript:

“Importantly, sites with GCN=7.5 are minority sites on the Ru NPs, with a low contribution to the overall TOF (see below).”

For the uncertainty of scaling relations, we have performed a sensitivity analysis on the slope of linear scaling relations on Ru NP, as shown in Response 1. According to the sensitivity analysis, the changes in the slope of linear scaling relations result in minor differences in the TOF.

3, the linear scaling relations for reaction energies and activation barriers with respect to strain are obtained on Ru extended surfaces. Considering the different atomic coordination numbers on Ru NP, the reliability of these relations is questionable to predict the reaction energies and activation barriers.

Response 3: The scaling relations with and without strain are obtained using extended surfaces that capture sites with different coordination. The use of scaling relations relies on the fact that electronic size effects in NP are negligible for sizes above about 2 nm. This was shown, for example, in J. Phys. Chem. Lett. 4, 222–226 (2013) where the d-states of Pt NPs were shown to be converged to the values of the extended surfaces for particles containing about 200 atoms.

The scaling relations with respect to strain are obtained using Ru(0001) and a stepped Ru(0001) modeling the B₅ site. As shown in Figure R2, the scaling with respect to strain is similar for the two cases. This result motivated us to use an averaged scaling relation for all sites.

Figure R2: Scaling relations of reaction energies and barriers with respect to strain.

We also explicitly addressed the question of whether the scaling relations for surfaces are applicable for NPs by comparing the effect of strain on Ru(0001) and a terrace on a Ru₃₈₀ NP, Supplementary Figure 9. The effect of strain is very similar in the two cases.

To verify whether the averaging over different sites in the scaling relation for strain affects the results, we compared two cases for Ru₇₈₀ NP. Figure R3a shows the particle as treated in the manuscript, thus with the same scaling relations used for all types of sites. Figure R3b shows a model that distinguishes between terrace and step sites. The energies for terrace sites on the NP are treated as strained Ru(0001) sites, whereas the energies for step sites are treated as strained stepped Ru(0001) sites. The TOF for the systems in Figure R3a

Editorial Note: Figure R4 in this Peer Review File is reproduced from Nilsson Pingel, T., Jørgensen, M., Yankovich, A.B., Grønbeck, H., Olsson, E. Influence of atomic site-specific strain on catalytic activity of supported nanoparticles. Nat. Commun. 2018, 9, 2722. Published under a Creative Commons Attribution 4.0 International License - <https://creativecommons.org/licenses/by/4.0/>.

and Figure R3b are calculated to be $3.3 \times 10^{-4} \text{ site}^{-1} \text{ s}^{-1}$ in both cases. The kinetic results demonstrate that averaging or distinguishing the sites with respect to strain has negligible influence on the reaction kinetics.

Figure R3: Ru₇₈₀ NPs that (a) distinguish and (b) not distinguish the step and terrace sites in the effects of strain.

4, how to define the strain on NP? especially for homogeneously strain? Duo the different atomic coordination numbers on Ru NP, the bond lengths are different on NP.

Response 4: We have defined strain in the NPs with respect to Ru bulk. The determination of strain for NP is more elaborate experimentally, where one generally uses an internal reference, *e.g.*, the center of a NP facet. The development of high-resolution transmission electron microscopy (HRTEM) provides an opportunity to quantitatively measure the strain distributions in NPs. In Nat. Commun., 9, 2722 (2018), the strain distribution in Pt NPs was measured and we reproduce some of the results in Figure R4.

Figure R4: Strain distribution in one NP measured by HRTEM [Nat. Commun., 9, 2722 (2018)].

The measured strain is typically distributed between -5% and 5%. We use this result to use a σ value of 0.05 in the simulations. The inherent strain can be introduced through strong metal-support interactions. When considering the support effect, the experimental

observations show that the strain at the interface ranges from 12% to 4%. In the simulations, we set the σ value to 0.10 and 0.15, to account for the strain effect induced by the strong metal-support interactions.

To clarify the definition of strain the NP, we have revised the text in manuscript on page 9:

“To simulate the kinetics over Ru NPs with a distribution of strained atoms, a random normal distribution is applied:

$$f(\Delta s) = \frac{1}{\sigma\sqrt{2\pi}} e^{-\frac{1}{2}\left(\frac{\Delta s - \bar{\Delta s}}{\sigma}\right)^2}$$

Where Δs is the strain, which is defined by the nearest neighbor difference with respect to bulk, ...”

Response to Reviewer 2:

The manuscript reports a theoretical study of ammonia synthesis on Ru nanoparticles. The study reveals a few important theoretical observations, especially the strain-TOF relation. The strain-TOF relation is due to kinetic couplings between strain and catalytic site contribution. The strain-TOF relation is independent of particle size. In the study, the authors mainly use kinetic Monte Carlo simulations based on potential energy surfaces fitted to density functional theory calculations. I agree that these theoretical insights offer strong support for the design of new catalysts. Given that ammonia synthesis is such an important and widely concerned topic, I believe the study is timely and suitable for the journal. The manuscript is well written. However, before I can recommend acceptance, the following points should be addressed to ensure that the theoretical methods and analyses are sound.

1. An important assumption applied in the study is the linear relations between the defined GCN and reaction energy. Can the authors justify why such linear relations should apply? As observed in Fig. 1a, there are outliers. The authors have discussed the outlier indicated by the solid marker, and have claimed (line 151) that it does not affect the kinetics simulation because the corresponding surface is not abundant. I have a few more questions related to this. Can the authors provide a more quantitative assessment of the impact of the outlier on the final results? Can the authors provide some insight into why the linear relation breaks significantly for this particular point? Can the authors check whether the discrepancy has something to do with the BEEF-vdW DFT functional used (or other settings, which are less likely)?

Response 1: We thank the Reviewer for the questions regarding the applicability and quality of the scaling relations, which have motivated us to do a thorough sensitivity analysis. Linear scaling relations for adsorbates on transition metal surfaces have been developed from DFT-calculations since the mid 90-ties are based on the fact that the chemical properties of metal atoms scale with the atom coordination and the local d-band center. However, there are, in some cases, details in the bond between the adsorbate and the metal atoms that are not captured by the scaling relations, giving rise to outliers. One outlier is the adsorption energy of N on Ru(0001).

The presence of the N-outlier for Ru(0001) means that the TOF for Ru(0001) is overestimated using the scaling relations. For Ru(0001), the TOF with the energy parameters obtained by linear scaling relations and by explicit DFT potential energy surfaces are $3.7 \times 10^{-6} \text{ site}^{-1} \text{ s}^{-1}$ and $6.3 \times 10^{-9} \text{ site}^{-1} \text{ s}^{-1}$, respectively. However, the overestimation for the Ru(0001) surface does not affect the kinetics on surfaces with Ru steps and Ru NPs. The reason that the kinetics are insensitive to the N adsorption energy of Ru(0001) is that the TOF on Ru(0001) surface, also with the scaling relation, is considerably smaller than that on step sites. Moreover, the Ru(0001) sites are minority sites on Ru NPs.

To further verify that the N-outlier on Ru(0001) does not affect the kinetics on Ru step surface and Ru NPs, we have performed new kMC simulations on both stepped Ru(0001) surface and Ru₇₈₀ NP, using the explicit DFT-calculated N adsorption energy for Ru(0001) surface and the scaling relations. The comparison of the TOF for the different cases is shown in Table R3. The results are very similar, which again suggests that the outlier for N-adsorption on Ru(0001) does not influence the kinetics on stepped Ru(0001) surface or Ru NPs.

Table R3. TOF by different energy parameters on Ru(0001) surface (site⁻¹ s⁻¹).

	Linear scaling relations	Explicit DFT energy
stepped Ru(0001) surface	7.0×10^{-5}	6.7×10^{-5}
Ru ₇₈₀ NP	3.3×10^{-4}	3.0×10^{-4}

The results in Table R3 are added to the revised Supplementary Table 25 and a sentence on the comparison has been added on page 9 in the revised manuscript:

“(A comparison between the TOFs evaluated with scaling relations and explicit DFT-calculations for a Ru₇₈₀ NP is given in the Supplementary Table 25.)”

In the revised manuscript, we have added an analysis of how the slopes of the scaling relations affect the TOFs, see the response to Reviewer 1 (point 1).

The underlying reason why the N-adsorption energy on Ru(0001) is an outlier is connected to the special adsorption geometry. The deviation of the N adsorption energy on Ru(0001) from the scaling relation is not connected to the choice of exchange-correlation (xc) functional. We have verified this by doing comparative computations with PBE, PBE+D3 and RPBE, as shown in Table R4. For each xc- functional, the Ru(0001) surface with a GCN of 7.50 is an outlier. We have added the comparison with different DFT functionals in Supplementary Table 7.

Table R4. N-adsorption energy (eV) calculated with different xc-functionals.

surface	GCN	Adsorption site	xc-functional			
			BEEF-vdW	PBE	PBE+D3	RPBE
Ru B edge	5.55	hcp hollow	-0.95	-1.16	-1.30	-0.92
Ru(10 $\bar{1}2$)	6.00	4-fold hollow	-0.83	-1.08	-1.28	-0.80
Ru(2 $\bar{1}\bar{1}2$)	6.50	hcp hollow	-0.70	-0.99	-1.11	-0.66
Ru(10 $\bar{1}1$)	6.96	4-fold hollow	-0.84	-1.07	-1.27	-0.78
Ru(10 $\bar{1}0$)	7.00	hcp hollow	-0.58	-0.79	-0.99	-0.51
Ru(0001)	7.50	hcp hollow	-0.98	-1.16	-1.33	-0.89
Ru D step	8.27	hcp hollow	-0.38	-0.56	-1.02	-0.27
Ru B step	8.72	hcp hollow	-0.49	-0.67	-0.89	-0.38
Ru A step	9.13	hcp hollow	-0.35	-0.32	-0.53	-0.03

To investigate the nature of N-adsorption on Ru(0001) further, we compared the structures of N-adsorption on Ru(0001) and Ru(10 $\bar{1}0$). The adsorption energy of Ru(10 $\bar{1}0$) follows the linear scaling relation, whereas the value on Ru(0001) is an outlier. Although the GCN and adsorption configuration are similar on the two surfaces, the N adsorption energies on these two surfaces are different by 0.40 eV.

An indication that a different component to the bonding is present for N on Ru(0001) is an analysis of the adsorption energy with respect to the d-band center of the surface atoms. The results are shown in Figure R5. There is a clear linear relation between the d-band center and N adsorption energy for all surfaces except Ru(0001).

**Figure R5:** Linear relation between N adsorption energy and d-band center of Ru at the site of adsorption.

For the N adsorption on Ru(0001) (left in Figure R6), the bond length between N atom and the neighboring Ru atoms on the surface and subsurface layer are 1.92 Å and 3.17 Å, respectively. On Ru(10 $\bar{1}0$) (right in Figure R6), the bond length between N atom and the neighboring Ru atoms on the surface layer is 1.92 Å, which is similar to that on Ru(0001).

However, the distance between N and the subsurface is 3.32 Å, which is significantly longer than that on Ru(0001). The shorter distance between N atom and the subsurface Ru atom on Ru(0001) than Ru(10 $\bar{1}$ 0) implies that the subsurface Ru atom may contribute to a stronger N adsorption energy.

Table R5. Comparison of N adsorption on Ru(0001) and Ru(10 $\bar{1}$ 0).

	Ru(0001)	Ru(10 $\bar{1}$ 0)
GCN	7.50	7.00
N adsorption site	3-fold hcp hollow	3-fold hcp hollow
N adsorption energy (eV)	-0.98	-0.58

Figure R6: N adsorption configuration on Ru(0001) and Ru(10 $\bar{1}$ 0). Atomic color code: N (blue), Ru surface layer (light gray), and Ru subsurface layer (dark gray).

The fact that the subsurface atom contributes to the bonding of N is further corroborated by a projected density of states (p-DOS) analysis, as shown in Figure R7. On both Ru(0001) and Ru(10 $\bar{1}$ 0) surfaces, the 2p-orbital of N atom forms the bonds with the 4d-orbitals of neighboring Ru atoms on the surface layer. In addition, the 2p_z orbital of N atom is bonded to the 4d_{z²} orbital of Ru atoms in the subsurface layer. This indicates that the Ru atom in the subsurface layer contributes to the N adsorption energy.

Figure R7: Projected density of states (p-DOS) analysis of N and Ru atoms on Ru(0001) and Ru(10 $\bar{1}$ 0).

2. The authors use a random strain model, which allows them to obtain TOF in the regime of experimental results. Can the authors justify why a random strain model is physically reasonable? My intuition is that the strain field in nanoparticles should be homogeneous or some continuously distributed field.

Response 2: The use of a random normal distribution is based on the experimentally observed strain distribution of NP surface, Nat. Commun., 9, 2722 (2018). These experiments show that the strain distributions across different regions of the NP surface appear random, with the strain ranging from -5% to 5%. Thus, we set the σ value to 0.05 to represent the inherent strain on the surface of NP. Moreover, the experiments showed that strain in the NPs can be induced by strong metal-support interactions. When considering the support effects, the strain at the interface is distributed from -12% to 4%. To account for the effect of strong metal-support interactions on the strain, we set the σ to be 0.10 and 0.15.

To further investigate the influence of specific strain configurations on the kinetics of NPs, we performed the kMC simulations on 256 randomly strained Ru NPs with σ values of 0.05, 0.1 and 0.15, as shown in Figure 2b in the manuscript. In Figure 2b, the standard deviation of TOF on 256 randomly strained Ru NPs at σ of 0.05, 0.10 and 0.15 are 0.00079,

0.0016 and 0.0021, respectively. The relatively narrow TOF distributions suggest that the kinetics on NPs are not highly sensitive to details in the strain distribution. Thus, we conclude that it is reasonable to use random strain distributions when exploring strain effect in NPs.

3. The authors claimed agreement with experiment, but based on my reading the agreement is only that the calculated TOF is in the same order of magnitude as experimental measurements. If I did not miss other information, then I would suggest the authors make the statements clearer. Given the agreement comes along with a few assumptions employed (linear relations and random strain model), one should be careful not to over-interpret the importance of “theory-experiment” agreement.

Response 3: We agree with the Reviewer that the statement regarding the agreement could have been more precise. In our simulations, both the TOF and the apparent activation energy on randomly strained ($\sigma=0.05$) Ru₇₈₀ NP fall within the reported experimental ranges. To clarify this, we have rephrased the statement on page 10, which now reads:

“The calculated TOF and apparent activation energy are within the range of experimental reports, which indicates that the kinetic model properly describes the experimental situation and consolidates the importance of strain effects”

4. In Fig. 1a and Fig. 2a, the data points are not clearly indicated in the caption, they are mentioned in the main text and SI table, I suggest the authors add more description sentences in the captions to improve the readability of the figures alone, where they can also refer to corresponding information in the SI explicitly.

Response 4: We thank the Reviewer for this comment. We have added the TOF-data with respect to strain (Figure 2a) to Supplementary Table 24. Moreover, we have modified the text in the caption of Figure 1, which now reads:

“Figure 1: (a) Adsorption energies versus GCN for each surface species (The data points are shown in Supplementary Tables 1-6.) Inset: energy barriers of N₂ dissociation step versus reaction energy. (b) Potential energy surfaces of sites with GCN of 9, 6.5 and 4. Inset: the Ru₇₈₀ NP colored according to the GCN of atop sites.”

5. Figure 1b depicts the potential energy surfaces of sites with varying GCN. Is it possible, from either an energetic or KMC perspective, for two NH₃ molecules to form simultaneously (e.g., NH₂* + NH* + 3H*)? It would benefit the authors to explicitly state whether these potential energy surfaces represent a reasonable simplification or if they merely illustrate one of several possible reaction pathways.

Response 5: The potential energy diagram represents the energetics for one complete catalytic cycle. With a high coverage of intermediates, the kinetics allow for the simultaneous

formation of two NH₃ molecules. This will indeed happen on large NPs. For example, on Ru₂₂₂₉₆ NP with a size of 8.99 nm, the N hydrogenation reaction ($N^* + H^* \rightarrow NH^* + *$), the NH hydrogenation reaction ($NH^* + H^* \rightarrow NH_2^* + *$) and the NH₂ hydrogenation reaction ($NH_2^* + H^* \rightarrow NH_3^* + *$) occur at the rates of 6.29×10^6 , 1.04×10^5 and 15 times per second, respectively. Thus, on average, about 15 NH₃ molecules are formed per second at different sites of Ru₂₂₂₉₆ NP.

The reaction pathway for ammonia synthesis on Ru NPs follows a dissociative reaction mechanism, where N₂ first dissociates into two N atoms, which subsequently undergo sequential hydrogenation to form gaseous NH₃. This mechanism has been widely validated and adopted in previous experimental (J. Catal. 192, 391-399, (2000), J. Vac. Sci. Technol. A, 1, 1247-1253 (1983)) and theoretical, Science, 307, 555-558, (2005) studies.

6. The statement in Line 141, “The difference in TOF between Ru(0001) and the stepped surface is only 20” suggests a difference in absolute value, which should be clarified.

Response 6: We thank the Reviewer for noting this typo. We have modified the sentence on page 8 to:

“The ratio between the TOFs for Ru(0001) and the stepped surface is only 20, which ...”

7. Considering strain in simulations is a significant contribution. The authors also employ linear relations to consider the strain effects on reaction energy and energy barrier. The linear parameters are fitted from DFT calculations, but the data for fit should be shown.

Response 7: We agree that the data for the strain should be reported in a clearer way. The parameters for the linear scaling relations with respect to strain are shown in Supplementary Table 18 and the underlying data is reported in Supplementary Tables 19 and 20. To make the linear scaling relations easier to read, we have in the revised SI adding Supplementary Figure 8 for linear scaling relations on Ru(0001) and stepped Ru(0001) surfaces for each elementary reaction.

8. Additionally, can the authors comment on the statistical error when they use a normal distribution to model randomly strained atoms.

Response 8: In Figure 2b, the standard deviation of TOF on 256 randomly strained Ru NPs at σ of 0.05, 0.10 and 0.15 are 0.00079, 0.0016 and 0.0021, respectively. We have added the statistical error in the manuscript on page 10, which now reads:

“Comparing the distributions at different σ values, the average TOFs at σ of 0.05, 0.10 and 0.15 are calculated to be $3.6 \times 10^{-3} \text{ site}^{-1} \text{ s}^{-1}$, $1.2 \times 10^{-2} \text{ site}^{-1} \text{ s}^{-1}$ and $1.9 \times 10^{-2} \text{ site}^{-1} \text{ s}^{-1}$, with the standard deviation of 0.00079, 0.0016 and 0.0021, respectively.”

9. In Figure 3(b), are the columns for N₂ dissociation (GCN=4) and NH₂ hydrogenation (GCN=8, strain=0) omitted due to their negligible values?

Response 9: Figure 3a shows that the N₂ dissociation step mainly occurs on the high GCN sites, whereas the NH₂ hydrogenation step mainly occurs on the low GCN sites. Thus, the contributions of N₂ dissociation step on GCN=4 sites and the NH₂ hydrogenation step on GCN=8 sites to DRC are negligible. We have added this information in the figure caption to Figure 3:

The contribution to the DRC from N₂ dissociation on GCN=4 sites is negligible and have been omitted. The NH₂ hydrogenation on GCN=8 is small and not visible.

10. Planck constant h in equation 6 and 7 is not mentioned.

Response 10: We thank the Reviewer for noting the omission. In the revised version of the manuscript, we mention the introduction of Planck's constant h in connection to Equation 7:

Where ΔG^{act} is the Gibbs free energy barrier, ΔS^{act} is the entropy barrier, ΔE^{act} is the energy barrier, and h is Planck's constant.”

During the revision, we found a typo in equilibrium constant (Equation 6). We have modified this error, which is now read as:

$$K^{ads} = \frac{k^{ads}}{k^{des}} = \exp\left(-\frac{\Delta G^{ads}}{k_B T}\right) \approx \exp\left(-\frac{\Delta E^{ads} - T\Delta S^{ads}}{k_B T}\right)$$

11. It is not clear what the authors mean by “break the limitation of linear scaling relations” in the abstract and conclusions.

Response 11: The constraint of linear scaling relations and the which to escape the scaling relations is often discussed in catalysis, see e.g. Nature Catalysis 2, 971-976 (2019). We mention the limitations of linear scaling relations in the Abstract and at the end of the Introduction. Kinetic analysis based on linear scaling relations considers generally only one type of sites. As adsorption energies and reaction barriers are correlated, the kinetic possibilities are limited for such types of systems. Instead, when performing reactions over NP, different elementary steps may occur on different sites, thus, the limitations of the scaling relations are broken. We have added a reference to Nature Catalysis 2, 971-976 (2019) in the Introduction.

Response to Reviewer 3:

This manuscript investigates the problem of determining the kinetics of ammonia synthesis over Ru nanoparticles. This continues to be a hot topic for research due to its industrial applications but also as a prototype for benchmarking catalytic models. In general, I find the methodology mostly sound and results useful. However, for a high impact journal such as Nat. Commun., I find some limitations that I would like more clarity on. Should these limitations be appropriately addressed, I would recommend publication in Nat. Commun., and believe the results will be of great interest to the catalysis community, including both experimental and computational researchers.

1, The work presented uses similar methodology to that presented by some of the same authors for CO oxidation over Pt nanoparticles (DOI: 10.1002/anie.201802113) and thus lacks some novelty due to this method already largely being presented. The present work represents a somewhat more complicated case, notably in the need for a more complex active site for N₂ dissociation, but also in the inclusion of strain consideration. The latter point is addressed very well but there is some discussion lacking about the effects of an active site that involves more than one surface atom. Can the authors contrast the present case with that of CO oxidation in order to make the methodology more generally applicable and/or allow the reader to understand what differences may arise for different types of reactions?

Response 1: We thank the Reviewer for the comment. It is true that we used the scaling relation Monte Carlo method for CO oxidation in previous publications. However, we think that the understanding of the kinetics of catalytic reactions over NPs is still emerging. To bridge the “materials gap” between structurally simple model system as single crystal surfaces and technical catalysts with supported NP, the first step is to account for the complex potential energy surface together with the kinetics couplings between different sites. The couplings for CO oxidation over Pt were simple in the sense that it mainly connected to the diffusion of O atoms to the edges, which are populated by CO at high temperatures. The situation is more complex for NH₃ synthesis where different reactions require different types of sites. N₂ dissociation requires one type of site (GCN>8), whereas hydrogenation requires another type of site (GCN<8). Furthermore, an additional novelty is that we are considering the effect of inherent strain on Wulff NPs, which is another important step to bridge the materials gap. Our results demonstrate that inclusion of kinetic coupling and inherent strain effects on the NPs is required to obtain quantitative agreement with experiments. To make this clearer, we modified the text on page 14 in the manuscript to:

“The result clearly shows that different sites are important for the NH₃ synthesis reaction, making it crucial to consider the assembly of sites when analyzing and simulating NH₃ synthesis over Ru NP. It is the assembly of sites rather than one particular site that govern the activity.”

2. On the topic of active sites, I find the use of generalised coordination numbers rather interesting for more complex active sites. It is stated that the B5 and A5 sites have generalised coordination numbers of 8.75 and 9.17, respectively. Inspecting Fig. 1(b), where A5 is visible, it appears that these are defined by the atom within the “cavity” formed by the A5 site. However, N₂ dissociation does not utilise this cavity atom but rather the bridge sites of the other 4 atoms of the A5 site. Can the authors please clarify how GCN was used to characterise these more complex sites?

Response 2: We thank the Reviewer for this comment. While it is true that the N₂ dissociation step occurs on the bridge sites of 4 atoms in the A₅ step site, rather than the cavity atom, the presence of the cavity atoms stabilizes the transition state of N₂ dissociation. This stabilization leads to the low activation barrier of 1.06 eV for A₅ step sites. In comparison, on Ru(10 $\bar{1}$ 2) surface, although the transition state of N₂ dissociation is also on the bridge sites of 4 atoms, the absence of cavity atoms increases the activation barrier to 1.22 eV. The GCN becomes, in this way, a signature that a proper dissociation site is present.

3. I am intrigued by the differences noted between the explicitly calculated DFT energy surface for Ru(0001) and that derived from the linear scaling relations. These seem to be very large but are described nicely in the text and indeed the authors state that there are not many sites similar to Ru(0001) in the overall particles. However, it does beg the question of whether similar effects might be present for other sites. Have the authors investigated the differences between explicit and linear scaling results for any other sites? It seems that these differences mostly arise from the description of the N₂ dissociation barrier, which therefore may limit the number of additional calculations required but improve the accuracy of outcomes significantly.

Response 3: The N-adsorption energy is an outlier in the scaling relations. We discuss this in detail in response to Reviewer 2 (point 1). We have no indications of other outliers in our extensive database of DFT calculations. The high stability of N atoms on the Ru(0001) appears to be related to an attractive bond between N and the Ru atom in the subsurface layer, which can come close.

4. Have the authors considered that the dissociation of N₂ may be represented by a two-step process for many surfaces where first the N₂ molecule “lies down” on the surface and then dissociates? (See, for example: DOI: 10.1021/acs.jpcc.9b09563, Ref. 20 in main article.) It is possible that better results are obtained by creating linear scaling relations with this configuration of adsorbed N₂, than there where it binds on top of an atom.

Response 4: We performed calculations for N₂ adsorption on Ru(0001) and Ru B₅ step surface in two configurations, namely binding on the top site (top binding) and lying down (lying down). A comparison of N₂ adsorption energies for different adsorption configurations is provided in Table R6.

Table R6. DFT calculations of N₂ adsorption energy (eV).

	Ru(0001)	Ru B ₅ step
top binding	-0.55	-0.75
lying down	-0.10	-0.44

To investigate the impact of N₂ adsorption configurations on the kinetics, we performed the kMC simulations on the stepped Ru(0001) surface, which includes both Ru(0001) and Ru B₅ step sites. Using the N₂ adsorption energies from different configurations, the TOF values for the “top binding” and “lying down” configurations for one trajectory are calculated to be $3.8 \times 10^{-4} \text{ site}^{-1} \text{ s}^{-1}$ and $3.7 \times 10^{-4} \text{ site}^{-1} \text{ s}^{-1}$, respectively. The small difference is within the standard deviation of the kMC simulations. These results confirm that the different N₂ adsorption configurations have a negligible effect on the kinetics. The N₂ adsorption/desorption step [$N_2(g) + * \rightleftharpoons N_2^*$] is a quasi-equilibrium process, and N₂ coverages are low. Thus, the reaction rate is not affected by the energy of adsorbed N₂ state.

5. A key aspect of the kinetic coupling is diffusion between sites of low and high GCN. However, I find detail on the diffusion process lacking somewhat in the manuscript. Are these diffusions incorporated explicitly in the KMC modelling and, if so, how? It is likely that significant barriers to diffusion may exist between different sites and some indication of the magnitudes of these barriers would be appreciated.

Response 5: It has been reported [Angew. Chem. Int. Ed. 45, 7046-7049 (2006)] that the energy barrier of diffusion is about 0.12 times the respective adsorbate binding energies. Thus, the diffusion processes are typically fast with respect to the other reaction steps. To confirm that the diffusion barriers of surface species in this system are low, we performed the DFT calculations of N and NH diffusion barriers on Ru(0001) surface, which represent 0.64 eV and 0.62 eV, respectively. These diffusion barriers are significantly lower than the reaction energies. We have added this information in the revised manuscript on page 21 where we also mention how the diffusion barriers are handled:

The barriers for adsorbate diffusion are known to be a fraction (about 0.12) of the adsorption energy.⁵⁰ The diffusion rates are for all considered adsorbates considerably faster than the rates for N₂ dissociation and NH₂ hydrogenation. The diffusion rates has, therefore, been scaled-down, while maintaining the quasi-equilibrium.

6, Related to the above, was adsorption and desorption considered at all active sites and at all steps in the reaction?

Response 6: Yes, the adsorption and desorption of N_2 , H_2 and NH_3 molecules on every site of NP are taken into account. The N_2 , H_2 molecules mainly adsorb (and desorb) on facet sites with the GCN ranging in 6.0 to 7.5, and the NH_3 molecules mainly desorb on low GCN site ($GCN < 5$).

7, Has any consideration been given to not just “whole particle” distribution of active sites but more local distribution? Presumably smaller regions with diverse active sites would lead to higher activity than regions where the diverse sites were separated by rather large distances. Could this be tested in the model?

Response 7: The diffusion rates are much faster than the surface reactions, indicating that the distance between the active sites for different elementary reactions has a small effect on the kinetics. To confirm that diffusion rates do not influence the kinetics in this system, we performed kMC simulations on two model systems derived from Ru_{780} NP, as shown in Figure R8.

The N_2 dissociation step mainly occurs on the high-GCN (blue) sites, while the hydrogenation steps mainly occur on the low-GCN (yellow) sites. In Figure R9a, these different active sites are located within a small region. In contrast, in Figure R9b, we set the active sites for N_2 dissociation and NH_2 hydrogenation steps to be separated by a large distance. The TOF values obtained from kMC simulations for the structures in Figure R9a and Figure R9b are $0.021 \text{ site}^{-1} \text{ s}^{-1}$ and $0.022 \text{ site}^{-1} \text{ s}^{-1}$, respectively. These similar TOF results confirm that the distance between different active sites does not affect the kinetics.

Figure R9: Model systems picked up from the Ru_{780} NP. The blue and yellow balls represent the active sites of N_2 dissociation and NH_2 hydrogenation steps, respectively.

8, While I understand the use of several distinct particle morphologies to compare shape activities, the octahedron and cube (and to a lesser extent the other motifs) do not represent very realistic particle shapes and indeed represent quite “extreme” points to study site distributions. Can the authors comment on whether slightly different Wulff constructions may be obtained (or perhaps at slightly different particle sizes) that might represent more realistic particles but differ in their site distributions?

Response 8: We evaluated different Wulff construction NPs within the size range of 5.5 nm to 6.0 nm, as shown in Figure R9. Given that the rate limiting N₂ dissociation step mainly occurs on the high GCN sites, the TOF of Ru NP is affected by the proportion of the high GCN sites. Figure R9 shows that even a slight change of size leads to different GCN distributions, which in turn results in significant difference in TOF. For example, on Ru₆₄₉₈ NP with the size of 5.96 nm, the ratio of high GCN (GCN>8.0) sites is 10%, yielding a TOF of $3.6 \times 10^{-3} \text{ site}^{-1} \text{ s}^{-1}$. However, for Ru₆₉₇₂ NP with the size of 5.96 nm, the ratio of high GCN is 26%, leading to a TOF of $5.5 \times 10^{-3} \text{ site}^{-1} \text{ s}^{-1}$. In the manuscript, this is visible in Figure 3e, which shows a large difference in TOF for particles of similar size.

Figure R9: GCN distributions of Ru NPs within the size range of 5.5 nm to 6.0 nm.

9, The study of strain is a very interesting aspect of the present study and, as the authors point out, is a factor often neglected in computational studies. The authors state that strain is usually present in technical catalysts and used as a motivation for its study here. I find this a very nice incorporation in the model. However, I wonder if the authors have made any attempt to quantify the degree of strain in the experiments and whether the incorporated strain is reasonable for the specific experimental results to which they compare? Specifically, the TOF is made to fall within the range of experiments (p9) by the incorporation of a certain amount of strain but this itself is not entirely convincing if the degree of strain in the experiments is not known. It could otherwise result in the “right answer for the wrong reason”, so to speak. Note that I am not suggesting that the incorporation of strain is not appropriate, it is just the firmness with which the comparisons with experiment are made and related to strain that I find slightly concerning.

Response 9: The assumption of random normal distribution is based on the experimentally observed strain distribution of NP surface [Nat. Commun. 9, 2722, (2018)]. These experiments show that the strain distributions across different regions of the NP surface are random, with the strain mainly ranging from -5% to 5%. Thus, we set the σ value to 0.05 to represent the inherent strain on the surface of NP. Moreover, the experiments showed that the strain on the NPs can be induced by the supports with strong metal-support interactions. When considering the support effects, the strain at the interface is distributed from -12% to 4%. To account for the effect of supports on the strain, we increase the σ to 0.10 and 0.15.

To further investigate the influence of specific strain configurations on the kinetics of NPs, we performed the kMC simulations on 256 randomly strained Ru NPs with σ values of 0.05, 0.1 and 0.15, as shown in Figure 2b. The relatively narrow TOF distributions suggest that the kinetics on NPs are not highly sensitive to the strain configuration. Thus, using the random strain model to explore the strain effect of NP is physically reasonable.

In terms of the homogeneous strain, the strain ratio is chosen from the experiments on different core-shell Ru alloys [J. Am. Chem. Soc. 142, 7036–7046, (2020), J. Am. Chem. Soc. 144, 19619-19626, (2022), J. Am. Chem. Soc. 145, 5710-5717, (2023)]. In these experiments, the homogenous strain of Ru NP ranges between -4% to 12%.

10, The BEP relations in Fig. S1 do not fit particularly well. How does this affect the overall outcome of the simulations? The blue line (*N + *H -> *NH) seems to be particularly anchored by the point at $\Delta E = \sim 0.15$ eV. Can the authors comment on how sensitive the TOF are to the goodness of these fits?

Response 10: We thank the Reviewer for this comment. We have performed a sensitivity analysis on the slope of linear scaling relations on Ru NPs. As the activation barriers are derived from scaling relations with respect to reaction energy, this sensitivity analysis also captures the uncertainty of activation barriers. The details of sensitivity analysis on linear scaling relations are provided in our response to Reviewer 1 (point 1). According to the sensitivity analysis, the changes in the slope of linear scaling relations for each hydrogenation step have a small effect on the TOF.

11, Fig. 1b is a little hard to read where the various labels point to. The authors could consider some tick marks on the horizontal axis to aid readability.

Response 11: We thank the Reviewer for this suggestion, and we have modified the tick marks of Figure 1b.

Please find below my report on the manuscript “Inherent strain and kinetic coupling determine the kinetics of ammonia synthesis over Ru nanoparticles”, written by Y. Yang *et al.*

This manuscript investigates the problem of determining the kinetics of ammonia synthesis over Ru nanoparticles. This continues to be a hot topic for research due to its industrial applications but also as a prototype for benchmarking catalytic models. In general, I find the methodology mostly sound and results useful. However, for a high-impact journal such as *Nat. Commun.*, I find some limitations that I would like more clarity on. Should these limitations be appropriately addressed, I would recommend publication in *Nat. Commun.*, and believe the results will be of great interest to the catalysis community, including both experimental and computational researchers.

1. The work presented uses similar methodology to that presented by some of the same authors for CO oxidation over Pt nanoparticles (*DOI: 10.1002/anie.201802113*) and thus lacks some novelty due to this method already largely being presented. The present work represents a somewhat more complicated case, notably in the need for a more complex active site for N₂ dissociation, but also in the inclusion of strain consideration. The latter point is addressed very well but there is some discussion lacking about the effects of an active site that involves more than one surface atom. Can the authors contrast the present case with that of CO oxidation in order to make the methodology more generally applicable and/or allow the reader to understand what differences may arise for different types of reactions?
2. On the topic of active sites, I find the use of generalised coordination numbers rather interesting for more complex active sites. It is stated that the B₅ and A₅ sites have generalised coordination numbers of 8.75 and 9.17, respectively. Inspecting Fig. 1(b), where A₅ is visible, it appears that these are defined by the atom within the “cavity” formed by the A₅ site. However, N₂ dissociation does not utilise this cavity atom but rather the bridge sites of the other 4 atoms of the A₅ site. Can the authors please clarify how GCN was used to characterise these more complex sites?
3. I am intrigued by the differences noted between the explicitly calculated DFT energy surface for Ru(0001) and that derived from the linear scaling relations. These seem to be very large but are described nicely in the text and indeed the authors state that there are not many sites similar to Ru(0001) in the overall particles. However, it does beg the question of whether similar effects might be present for other sites. Have the authors investigated the differences between explicit and linear scaling results for any other sites? It seems that these differences mostly arise from the description of the N₂ dissociation barrier, which therefore may limit the number of additional calculations required but improve the accuracy of outcomes significantly.
4. Have the authors considered that the dissociation of N₂ may be represented by a two-step process for many surfaces where first the N₂ molecule “lies down” on the surface and then dissociates? (See, for example: *DOI:*

10.1021/acs.jpcc.9b09563, Ref. 20 in main article.) It is possible that better results are obtained by creating linear scaling relations with this configuration of adsorbed N₂, than there where it binds on top of an atom.

5. A key aspect of the kinetic coupling is diffusion between sites of low and high GCN. However, I find detail on the diffusion process lacking somewhat in the manuscript. Are these diffusions incorporated explicitly in the KMC modelling and, if so, how? It is likely that significant barriers to diffusion may exist between different sites and some indication of the magnitudes of these barriers would be appreciated.
6. Related to the above, was adsorption and desorption considered at all active sites and at all steps in the reaction?
7. Has any consideration been given to not just “whole particle” distribution of active sites but more local distribution? Presumably smaller regions with diverse active sites would lead to higher activity than regions where the diverse sites were separated by rather large distances. Could this be tested in the model?
8. While I understand the use of several distinct particle morphologies to compare shape activities, the octahedron and cube (and to a lesser extent the other motifs) do not represent very realistic particle shapes and indeed represent quite “extreme” points to study site distributions. Can the authors comment on whether slightly different Wulff constructions may be obtained (or perhaps at slightly different particle sizes) that might represent more realistic particles but differ in their site distributions?
9. The study of strain is a very interesting aspect of the present study and, as the authors point out, is a factor often neglected in computational studies. The authors state that strain is usually present in technical catalysts and used as a motivation for its study here. I find this a very nice incorporation in the model. However, I wonder if the authors have made any attempt to quantify the degree of strain in the experiments and whether the incorporated strain is reasonable for the specific experimental results to which they compare? Specifically, the TOF is made to fall within the range of experiments (p9) by the incorporation of a certain amount of strain but this itself is not entirely convincing if the degree of strain in the experiments is not known. It could otherwise result in the “right answer for the wrong reason”, so to speak. Note that I am not suggesting that the incorporation of strain is not appropriate, it is just the firmness with which the comparisons with experiment are made and related to strain that I find slightly concerning.
10. The BEP relations in Fig. S1 do not fit particularly well. How does this affect the overall outcome of the simulations? The blue line (*N + *H -> *NH) seems to be particularly anchored by the point at $\Delta E = \sim 0.15$ eV. Can the authors comment on how sensitive the TOF are to the goodness of these fits?
11. Fig. 1b is a little hard to read where the various labels point to. The authors could consider some tick marks on the horizontal axis to aid readability.